# Shape Non-rigid Kinematics (SNK):
# A Zero-Shot Method for Non-Rigid Shape Matching via Unsupervised Functional Map Regularized Reconstruction

**Souhaib Attaiki**
LIX, École Polytechnique, IP Paris
attaiki@lix.polytechnique.fr

**Maks Ovsjanikov**
LIX, École Polytechnique, IP Paris
maks@lix.polytechnique.fr

## Abstract

We present Shape Non-rigid Kinematics (**SNK**), a novel zero-shot method for non-rigid shape matching that eliminates the need for extensive training or ground truth data. **SNK** operates on a single pair of shapes, and employs a reconstruction-based strategy using an encoder-decoder architecture, which deforms the source shape to closely match the target shape. During the process, an unsupervised functional map is predicted and converted into a point-to-point map, serving as a supervisory mechanism for the reconstruction. To aid in training, we have designed a new decoder architecture that generates smooth, realistic deformations. **SNK** demonstrates competitive results on traditional benchmarks, simplifying the shape-matching process without compromising accuracy. Our code can be found online: https://github.com/pvnieo/SNK.

## 1 Introduction

Shape matching, specifically non-rigid shape matching, is a fundamental problem in Computer Vision, Computational Geometry, and related fields. It plays a vital role in a wide range of applications, including statistical shape analysis [1, 2], deformation [3], registration [4], and texture transfer [5], among others. This problem involves finding a correspondence between two shapes that may differ due to non-rigid deformations, such as bending, stretching, or compression.

Traditional methods for addressing this problem have predominantly been axiomatic, offering several benefits, such as theoretical guarantees [6] and their application in different scenarios [7, 8, 9, 10]. Some of these methods include intrinsic isometries-based methods [11, 12], parametrization domain-based methods [13, 14], and functional map methods [15, 16, 17]. However, these methods often heavily rely on the availability of good initialization [18, 19, 20] or the use of handcrafted descriptors [21, 22, 23]. This dependence can be a limiting factor, constraining the performance of these methods and making them less effective in situations where these requirements cannot be adequately met.

Learning-based approaches have emerged as an alternative to these traditional methods, with notable examples including Deep Functional Maps methods [24, 25, 26, 27, 28, 29], dense semantic segmentation methods [30, 31, 32, 33], or template-based methods [34] which have shown significant promise in solving the shape-matching problem. The strength of these methods lies in their capacity to learn features and correlations directly from data, outperforming handcrafted descriptors. However, these techniques introduce a significant challenge. Even when the task is to match just a single pair of shapes, these methods necessitate the collection and training on a large data set, often in the tens of thousands [34]. The demand for these extensive data collections, coupled with the long training times that can extend into days [35, 36], can be particularly onerous.

37th Conference on Neural Information Processing Systems (NeurIPS 2023).

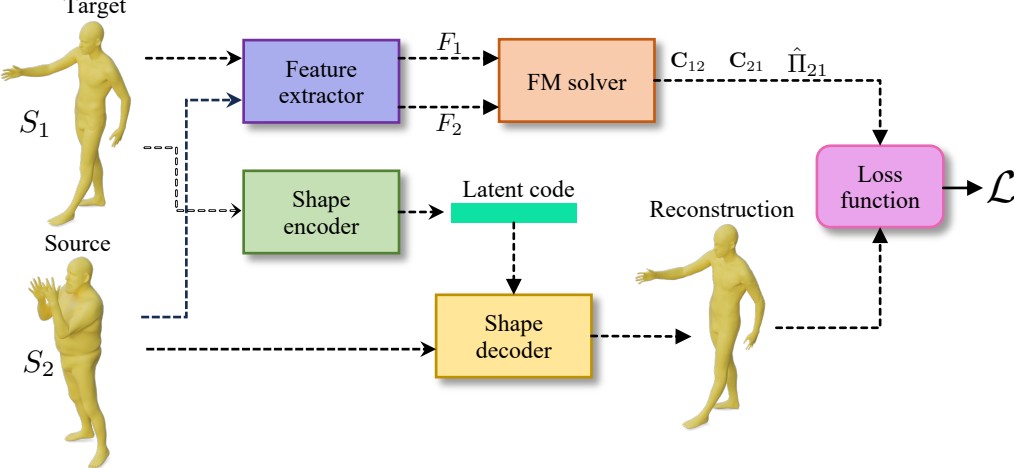

Figure 1: **Method overview**. Given two shapes as input, we deform a source shape to resemble the target. Our network can be trained using just a single pair of shapes and uses a combination of spatial and spectral losses, as well as a regularization on the decoder.

Our proposed method, Shape Non-rigid Kinematics (SNK), positions itself uniquely in this landscape, combining the benefits of both axiomatic and learning-based approaches while addressing their limitations.

In particular, SNK works directly on a given pair of shapes, without requiring any prior training. This removes the need for large data sets or pre-existing examples. Unlike axiomatic methods, however, that rely on fixed input features such as HKS or SHOT, we employ a deep neural network for feature computation. As we demonstrate below, this use of a neural network introduces a significant prior for our method, setting it apart from purely geometric approaches. We thus refer to our method as "zero-shot" to highlight its ability to be trained on a single pair of shapes, while using a neural network for feature learning.

SNK is applied to each pair individually using a reconstruction-based strategy that employs an encoder-decoder architecture. By doing so, SNK effectively deforms the source shape to closely match the target shape. A point-to-point map is then extracted by a simple nearest neighbor between the target and the deformed source.

At the heart of our method lies the functional map paradigm. In our approach, a functional map is generated in an unsupervised manner and then converted into a point-to-point map. This map is subsequently used as supervision for the decoder, ensuring the accuracy of the shape reconstruction process. To confine the deformation to plausible transformations, we introduce a new decoder architecture that is grounded on the PriMo energy [37]. Lastly, our method is trained using losses in both the spatial (reconstruction) and spectral (functional map) domains, ensuring comprehensive performance improvement.

Overall, our contributions can be summarized as follows:

- We have devised a new decoder architecture rooted in the PriMo energy concept. This architecture facilitates the production of deformations that appear smooth and natural.

- We have shown that a loss function, which imposes penalties on both spatial and spectral quantities, is adequate for deriving matches on a single pair of shapes without any prerequisite training.

- We have developed a method for zero-shot shape matching that attains state-of-the-art results among methods that are not trained on shape collections. Furthermore, it competes with, and frequently outperforms, several supervised training-based approaches.

## 2 Related Works

Non-rigid shape matching is a widely researched subject, and providing a comprehensive overview is beyond the scope of this paper. However, we will briefly discuss methods closely related to ours. For a more detailed exploration, we encourage interested readers to refer to recent surveys [38, 39, 40, 41].

**Functional Map**   Our method is built upon the functional map paradigm, which since its introduction [15], has been widely used for nonrigid shape matching [27, 19, 18, 17, 16, 7, 9, 42, 29, 43, 44, 45, 46]. It offers a distinct advantage by transforming the complexity of pointwise maps optimization, originally quadratic relative to the vertex count, into a more manageable process of optimizing a functional map (which consists of small quadratic matrices), making the optimization process feasible, see [47] for an overview.

In the pursuit of identifying the functional map, initial studies depended on handcrafted feature functions defined on both source and target shapes, such as HKS [22], WKS [23], and SHOT [21] features. Subsequent research enhanced this pipeline by incorporating extra regularization [16, 8, 48], adjusting it to accommodate partial shapes [9, 49, 10, 7], and suggesting efficient refinement techniques [19, 50, 51]. Contrarily, while these techniques often rely on an initial map for refinement or handcrafted features that lack robustness [25], our approach distinctively learns features by precisely fitting the source and target shapes using a neural network.

**Learning-based Methods for Shape Matching**   The field of shape matching has recently seen a surge in success with learning-based methods. These approaches take on various forms, including framing the issue as a segmentation problem, which involves training a network to segment each vertex on the source shape into a class corresponding to a vertex ID on a template shape [30, 31, 32, 33]. Other methods include template fitting, which deforms a template to align with source and target shapes [34], and the use of functional maps [24, 28, 52, 25].

The latter method involves a neural network that extracts features from source and target shapes to predict the functional map, which is then transformed into a point-to-point map. This innovative line of research was pioneered by FMNet [24] and subsequently expanded in numerous studies [27, 29, 25, 26]. These studies used ground-truth functional maps for supervision.

Simultaneously, a parallel line of research focuses on unsupervised learning, which proves to be beneficial when ground truth correspondences are unavailable. These methods ensure structural properties like bijectivity and orthonormality on functional maps in the reduced basis [28, 29], penalize the geodesic distortion of the predicted maps [53, 54], or harmonize intrinsic and extrinsic shape alignment [35, 55].

However, unlike axiomatic approaches, these methods are typically not competitive when applied on a single pair without prior training. In contrast, our method, though it is based on a neural network, can be directly applied to a single pair without any prerequisite training.

**Shape Reconstruction**   Our method employs an encoder-decoder framework to deform the source shape to fit the target shape. One significant challenge in this domain is crafting a shape decoder architecture capable of generating a 3D shape from a latent code. Early efforts in this field, such as those by Litany *et al.*, Ranjan *et al.*, and Bouritsas *et al.* [56, 57, 58], built on graph architecture, defining convolution, pooling, and unpooling operations within this structure. However, they presumed uniform triangulation across all shapes, rendering them ineffective for shape matching.

Other methods, such as Zhang *et al.* [59], predict occupancy probabilities on a 3D voxel grid, but the resource-intensive nature of dense, volumetric representations restricted resolution. Different approaches decod shapes as either raw point clouds [60, 61, 62] or implicit representations [63, 63]. Nevertheless, these methods overlooked the point ordering, making them unsuitable for matching.

In our work, we propose a novel decoding architecture, which leverages the architecture of DiffusionNet [64] as a foundational element. Our approach computes a rotation and a translation matrix for every face on the source shape to align with the target shape. While previous methods relied on energies such as As-Rigid-As-Possible (ARAP) [65], Laplacian [66], or edge contraction [34] to restrict the decoder, our approach is adept at using the PriMo energy [37]. This energy encourages smooth, natural deformations that effectively preserve local features, particularly in areas of bending [67], providing a more constrained and intuitive approach to shape deformation.

## 3 Notation & Background

Our work adopts the functional map paradigm [68] in conjunction with the PriMo energy [37]. In the following section, we briefly introduce these concepts, where we will use consistent notation throughout the paper.

### 3.1 Notation

Given a 3D shape $S_i$ composed of $n_i$ vertices and represented as a triangular mesh, we perform a cotangent Laplace-Beltrami decomposition [69] on it. The first $k$ eigenvalues of $S_i$ are captured in the matrix $\Phi_i \in \mathbb{R}^{n_i \times k}$. We also create a diagonal matrix $\Delta_i \in \mathbb{R}^{k \times k}$, where the diagonal elements hold the first $k$ eigenvalues of $S_i$.

Next, the diagonal matrix of area weights, $M_i \in R^{n \times n}$, is defined. It's worth mentioning that $\Phi_i$ is orthogonal in relation to $M_i$, and that $\Phi_i^\top M_i \Phi_i$ is equivalent to the $\mathbb{R}^{k \times k}$ identity matrix, denoted as $I_k$. Lastly, we represent $\Phi_i^\dagger = \Phi_i^\top M_i$ using the (left) Moore-Penrose pseudo-inverse symbol, $\cdot^\dagger$.

### 3.2 Functional Map Pipeline

Denoting source and target shapes as $S_1$ and $S_2$, we define a pointwise map, $T_{12} : S_1 \rightarrow S_2$, encapsulated in binary matrix $\Pi_{12} \in \mathbb{R}^{n_1 \times n_2}$. To manage its quadratic growth with vertex count, we employ the functional map paradigm [15], yielding a $(k \times k)$ size functional map $C_{21} = \Phi_1^\dagger \Pi_{12} \Phi_2$, with $k$ usually around 30, making the optimization process feasible.

Feature functions $F_1$ and $F_2$ are then derived from $S_1$ and $S_2$ respectively, with their spectral coefficients calculated as $\mathbf{A}_i = \Phi_i^\dagger F_i$. This enables the formulation of an optimization problem:

$$\underset{\mathbf{C}}{\arg\min} \|\mathbf{C}\mathbf{A}_1 - \mathbf{A}_2\|_F^2 + \lambda \|\mathbf{C}\Delta_1 - \Delta_2\mathbf{C}\|, \tag{1}$$

where $\mathbf{C}$ is the desired functional map and the second term is a regularization to promote structural correctness [25]. Finally, the point map $\Pi_{21} \in \{0,1\}$ is derived from the optimal $\mathbf{C}_{12}$, based on the relationship $\Phi_2\mathbf{C}_{12} \approx \Pi_{21}\Phi_1$, using methods such as nearest neighbor search or other post-processing techniques [18, 50].

### 3.3 PriMo Energy: Prism Based Modeling

The PriMo approach, introduced in [37], is a prominent method in shape deformation [67], employing a model that mimics the physically plausible behavior of thin shells and plates [70, 71]. In this model, a volumetric layer is formed by extruding the input mesh along vertex normals, thus generating rigid prisms for each mesh face (refer to Fig. 2). These prisms are interconnected by elastic joints, which stretch under deformation—the extent of this stretch effectively determines the deformation energy. Interestingly, this formulation permits a unique global minimum of the energy in the undeformed state, with any form of bending, shearing, twisting, or stretching causing an increase in this energy.

We elaborate on the elastic joint energy using the notation in Fig. 2. Two adjacent prisms, denoted $P_i$ and $P_j$, are considered. In their undeformed state, they share a common face that might no longer match after deformation. The face of $P_i$ adjacent to $P_j$ is defined as a rectangular bi-linear patch $\mathbf{f}^{i \rightarrow j}(u, v)$, which interpolates its four corner vertices. The corresponding opposite face is denoted as $\mathbf{f}^{j \rightarrow i}(u, v) \subset P_j$. The energy between $P_i$ and $P_j$ is thus defined as:

$$E_{ij} := \int_{[0,1]^2} \|\mathbf{f}^{i \rightarrow j}(u, v) - \mathbf{f}^{j \rightarrow i}(u, v)\|^2 du\, dv \tag{2}$$

$$E := \sum_{\{i,j\}} w_{ij} \cdot E_{ij}, \qquad w_{ij} := \frac{\|\mathbf{e}_{ij}\|_2^2}{|F_i| + |F_j|} \tag{3}$$

Here, $E$ is the deformation energy of the whole mesh, and the energy contribution from each pair $P_i, P_j$ is influenced by the areas of their corresponding mesh faces $F_i, F_j$, and the squared length of their shared edge $\mathbf{e}_{ij}$. In the original PriMo paper, users are allowed to dictate the positions and orientations of some prisms, while an optimization technique is deployed to minimize the total

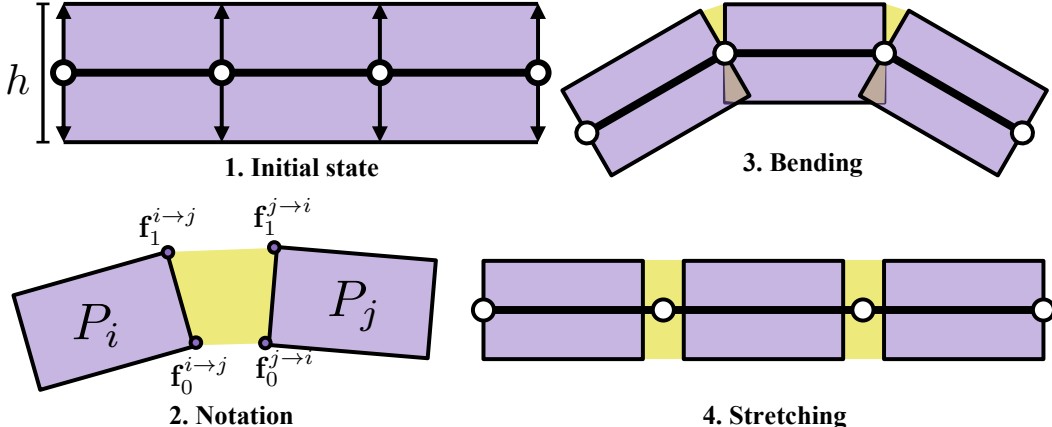

Figure 2: **PriMo construction**. The mesh vertices are represented as white circles. The elastic joints are colored yellow.

deformation energy. In contrast, in our approach, the deformation is the result of the decoder's output and the PriMo energy is utilized to regularize it.

# 4 Method

In this section, we delve into our proposed method for zero-shot matching, which consists of two primary modules. The initial (functional map) module, is responsible for predicting functional maps between the source and target shapes and converting these into point-to-point (p2p) maps. The second module, an encoder-decoder, encodes the target shape into a latent code. Subsequently, the decoder utilizes this latent code to morph the source shape to align with the target shape.

Diverging from previous studies, we incorporate losses in *both* the spatial and spectral domains. Spatially, we utilize the p2p map, as predicted by the functional map module, to ensure the reconstruction closely resembles the target. Spectrally, we enforce structural properties such as bijectivity and orthogonality on the functional map. Furthermore, to facilitate the learning process, we minimize the PriMo energy on the decoder, which ensures it generates a natural-looking and smooth deformation.

In the subsequent sections, we elaborate on each stage of our proposed model, and we will use the same notation as Sec. 3.

## 4.1 Functional Map Module

The functional map module is designed with a two-fold objective - to predict both fmap and p2p maps reciprocally between the source and target shapes. This module is composed of two key elements: the feature extractor and the fmap solver, as illustrated in Fig. 1.

The feature extractor's purpose is to extract features from the raw geometry that are robust to shape deformation and sampling, which are later employed for predicting the fmap. To achieve this, we use the DiffusionNet architecture [64], following the recent approaches in the domain [27, 52, 26, 72]. The feature extractor processes both the target and the source shapes, $S_1$ and $S_2$ respectively, and extracts pointwise features in a Siamese manner, denoted as $F_1$ and $F_2$.

Following the feature extraction, the fmap solver accepts the features $F_1$ and $F_2$, and predicts an initial approximation of the fmaps $\mathbf{C}_{12}^0$ and $\mathbf{C}_{21}^0$ utilizing Eq. (1). These fmaps are matrix solutions to the least square optimization, and thus do not necessarily originate from p2p maps. To rectify this, we draw inspiration from recent methodologies [43, 18, 45, 73, 74] and predict a new set of fmaps that stem from p2p maps. Initially, we transform the preliminary fmaps into an early estimation of p2p maps by applying:

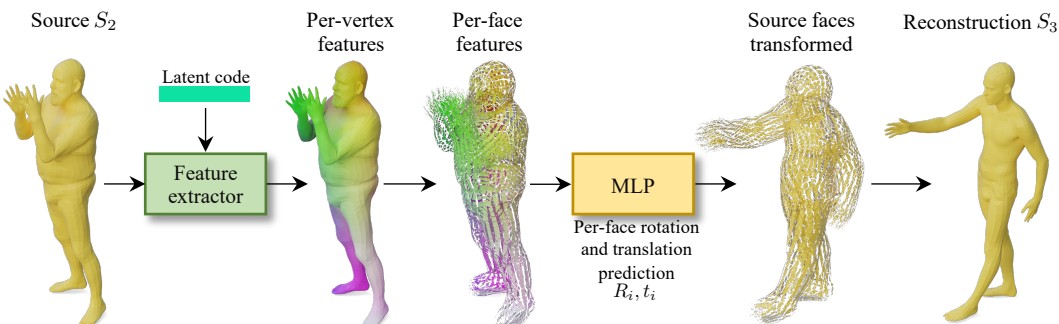

Figure 3: **Prism Decoder**. Our novel architecture, built upon DiffusionNet, initiates the process by extracting per-vertex features, which are subsequently consolidated into per-face features. These features are then processed by a Multilayer Perceptron (MLP) to generate per-face rotation and translation, which are used to rigidly transform the input faces. The transformed faces are then consolidated to produce the final reconstruction.

$$\hat{\Pi}_{21}^0(i, j) = \frac{\exp\left(\langle G_2^i, G_1^j\rangle/\tau\right)}{\sum_{k=1}^{n_1} \exp\left(\langle G_2^i, G_1^k\rangle/\tau\right)}. \tag{4}$$

Here, $G_1 = \Phi_1$, $G_2 = \Phi_2 \mathbf{C}_{12}^0$, $\hat{\Pi}_{21}^0$ is a soft p2p map predicted in a differentiable manner, $\langle\cdot,\cdot\rangle$ represents the scalar product, and $\tau$ denotes a temperature parameter. We predict $\hat{\Pi}_{12}^0$ in a comparable way, followed by predicting the final fmaps $\mathbf{C}_{12}$ and $\mathbf{C}_{21}$ that arise from p2p maps using the definition $\mathbf{C}_{ij} = \Phi_j^\dagger \hat{\Pi}_{ji}^0 \Phi_i$. Ultimately, we estimate the final p2p map $\hat{\Pi}_{ij}$ from $\mathbf{C}_{ji}$ by applying Eq. (4) for a second time. This procedure aligns with two iterations of the ZoomOut algorithm [18].

## 4.2 Shape Reconstruction Module

The goal of the shape reconstruction module is to deform the source shape to match the target shape. It is composed of a encoder that extract a latent code that summarize the target shape, and a decoder that takes the latent code as well as the source shape and deform it to match the target.

The encoder is a DiffusionNet network [64], that takes the 3D coordinates of the target shape, and produce a $d_1$-dimensional vertex-wise features. A max pooling is applied in the vertex dimension to obtain a latent code $l \in \mathbb{R}^{d_1}$.

**The prism decoder**  We introduce a novel architecture for the decoder, utilizing DiffusionNet at its core. An overview of this architecture is presented in Fig. 3. The decoder receives two inputs: the 3D coordinates of the source shape and the latent code $l$. The latent code is duplicated and combined with the XYZ coordinates, which are subsequently inputted into the DiffusionNet architecture. This process generates vertex-wise features $D \in \mathbb{R}^{n_2 \times d_2}$ of dimension $d_2$.

These vertex-wise features $D$ are then transformed into face-wise features $G \in \mathbb{R}^{f_2 \times d_2}$, where $f_2$ denotes the number of faces of the source shape. This transformation is achieved via mean pooling, where the three features corresponding to the vertices constituting each face are averaged: $G_i = \frac{1}{3}\sum_{j \in \mathbf{F}_i} D_j$, with $\mathbf{F}_i$ representing the $i^{th}$ face of the target shape.

These features are further refined by a compact network composed of pointwise linear layers and ReLU nonlinearities, producing outputs $H \in \mathbb{R}^{f_2 \times 12}$. The first three features are interpreted as a translation vector $t_i$, and the final nine features are reshaped into a $3 \times 3$ matrix $R_i^0$. This matrix is subsequently transformed into a rotation matrix $R_i$ by solving the Orthogonal Procrustes problem [75]:

$$R = \arg\min_\Omega \|\Omega - R^0\|_F \qquad \text{subject to} \quad \Omega^\top \Omega = I_3. \tag{5}$$

With $t_i$ and $R_i$ defined for each face, we transform the original source shape's faces $\mathbf{F}$, resulting in a new set of faces $\mathbf{F}'$. The vertices of the decoded shape are then constructed by averaging the new face coordinates' positions for all the faces they belong to.

## 4.3   Training Procedure

Our method operates in a zero-shot manner, meaning that it is trained on each new pair of shapes independently. This approach merges the benefits of axiomatic methods, which do not rely on large training sets and extended training periods, and the advantages of learning methods, which facilitate matching through *learned* features, transcending the limitations of handcrafted features used in axiomatic methods.

The process of our method unfolds as follows: the target and source shapes, designated $S_1$ and $S_2$, are processed through the functional map module. The output of this stage comprises the fmap and p2p maps, represented as $\mathbf{C}_{ij}$ and $\hat{\Pi}_{ij}$. Concurrently, a latent code, $l$, is extracted via the encoder and employed to deform the source shape to align with the target shape. The resulting deformed shape, $S_3$, maintains the same triangulation as the source shape. Utilizing all these outputs, a loss, denoted as $\mathcal{L}$, is computed, which subsequently guides the optimization of the parameters of the various networks through gradient descent. This process is iteratively executed until either the loss ceases to decrease or the maximum number of iterations is achieved. The output that yields the lowest loss is ultimately chosen for matching.

In this context, a p2p map between the source and target shape, known as $T_{21}^0$, is derived by performing a nearest neighbor search between the reconstructed shape $S_3$ and the target shape $S_1$. As suggested in previous works [27, 52, 25, 29, 64, 26], we refine this map by applying the ZoomOut algorithm [18]. This method enhances the map by iterating between the spectral and spatial domains, gradually augmenting the number of spectral basis functions. The refined map, $T_{21}$, is then employed for evaluation.

**Loss function**   In training our network, we employ multiple loss functions. Contrary to prior works [29, 28, 36, 35] that solely utilize either spatial or spectral losses, our method synergistically integrates both types for enhanced performance. The overall loss function is as follows:

$$\mathcal{L} = \lambda_{\mathrm{mse}}\mathcal{L}_{\mathrm{mse}} + \lambda_{\mathrm{fmap}}\mathcal{L}_{\mathrm{fmap}} + \lambda_{\mathrm{cycle}}\mathcal{L}_{\mathrm{cycle}} + \lambda_{\mathrm{primo}}\mathcal{L}_{\mathrm{primo}} \tag{6}$$

We denote $\mathbf{S}_i$ as the XYZ coordinates of shape $S_i$. The Mean Square Error (MSE) loss, $\mathcal{L}_{\mathrm{mse}}$, is employed to ensure that the reconstructed shape closely resembles the target shape using the predicted p2p map, it's calculated as $\mathcal{L}_{\mathrm{mse}} = \|\hat{\Pi}_{21}\mathbf{S}_1 - \mathbf{S}_3\|_F^2$.

The fmap loss, $\mathcal{L}_{\mathrm{fmap}}$, is used to regularize the structural properties of the predicted fmaps, such as bijectivity and orthogonality [28]: $\mathcal{L}_{\mathrm{fmap}} = \|\mathbf{C}_{ij}\mathbf{C}_{ji} - I_k\|_F^2 + \|\mathbf{C}_{ij}^\top\mathbf{C}_{ij} - I_k\|_F^2$. The bijectivity constraint ensures that the map from $S_i$ through $S_j$ and back to $S_i$ is the identity map, while the orthogonality constraint enforces local area preservation of the map.

The cycle loss, $\mathcal{L}_{\mathrm{cycle}}$, enforces cyclic consistency between predicted p2p maps: $\mathcal{L}_{\mathrm{cycle}} = \|\hat{\Pi}_{12}\hat{\Pi}_{21}\mathbf{S}_1 - \mathbf{S}_1\|_F^2$. Lastly, the primo energy, $\mathcal{L}_{\mathrm{primo}}$, is employed to encourage the decoder to yield a coherent shape and to ensure the deformation appears smooth and natural, facilitating the training process. It is calculated as the energy $E$ from Eq. (3).

## 5   Results

In this section, we evaluate our method on different challenging shape matching scenarios.

## 5.1   Datasets

Our experimental setup spans two tasks and involves four distinct datasets referenced in previous literature, incorporating both human and animal representations.

For near-isometric shape matching, we employed several standard benchmark datasets, namely, **FAUST** [2], **SCAPE** [76], and **SHREC**'19 [77]. To enhance the complexity, we selected the remeshed versions of these datasets, as suggested in [29, 25, 48], instead of their original forms. The **FAUST**

Table 1: **Quantitative Results on Near-Isometric Benchmarks**. The **best** and second-best results are highlighted. Our method comes out on top among axiomatic methods and matches or even surpasses some trained methods.

| | METHOD | FAUST | SCAPE | SHREC |
|---|---|---|---|---|
| *Axiom* | Ini + BCICP [19] | 6.4 | 11 | - |
| | Ini + ZoomOut [18] | 6.1 | 7.5 | - |
| | Smooth Shells [55] | 2.5 | 4.7 | 12.2 |
| | DiscreteOp [43] | 5.6 | 13.1 | - |
| *Sup* | 3D-CODED [34] | 2.5 | 31 | - |
| | FMNet [24] | 11 | 17 | - |
| | GeoFMNet [25] | 3.1 | 4.4 | 9.9 |
| | DiffGeoFMNet [64] | 2.6 | 2.9 | 8.5 |
| | TransMatch [80] | 2.7 | 18.3 | 14.5 |
| *Unup* | Unsup. FMNet [53] | 10.0 | 16.0 | - |
| | SurFMNet [28] | 15.0 | 12.0 | 30.1 |
| | Cyclic FMaps [54] | 14.8 | 12.7 | 36.5 |
| | WSupFMNet [29] | 3.3 | 7.3 | 11.3 |
| | Deep Shells [35] | **1.7** | **2.5** | 21.1 |
| | NeuroMorph [36] | 8.5 | 29.9 | 26.3 |
| | **SNK** (Ours) | 1.8 | 4.7 | **5.8** |

dataset contains 100 shapes, depicting 10 individuals in 10 different poses each, with our evaluation focused on the last 20 shapes. The SCAPE dataset comprises 71 distinct poses of a single person, and we based our assessment on the final 20 shapes. The SHREC dataset, with its notable variations in mesh connectivity and poses, presents a more significant challenge. It consists of 44 shapes and a total of 430 evaluation pairs. For unsupervised methods, we used the oriented versions of the datasets as detailed in [29]. Moreover, for training methods, we showcase their optimal performance on SHREC using a model trained either on FAUST or SCAPE.

In the context of non-isometric shape matching, we conducted our experiments using the SMAL animal-based dataset [78, 79], hereinafter referred to as SMAL. This dataset includes 50 distinct, non-rigid 3D mesh models of organic forms. We divided these into two equal sets of 25 for training and testing, ensuring that the specific animals and their poses used for testing were not seen during the training phase.

## 5.2 Near-Isometric Shape Matching

In this section, we evaluate our method's performance in near-isometric, non-rigid shape matching. We draw comparisons with several contemporary approaches in the field of non-rigid shape matching. These can be broadly divided into three categories. The first category is axiomatic approaches and it includes BCICP [48], ZoomOut [18], Smooth Shells [55], and DiscreteOp [43]. The second category is supervised learning approaches and it comprises methods such as 3D-CODED [34], FMNet [24], GeomFmaps [25], DiffGeoFMNet [64], and TransMatch [80]. Finally the third category is unsupervised learning approaches and it includes Unsupervised FMNet [53], SURFMNet [28], Cyclic FMaps [81], Weakly Supervised FMNet [29], DeepShells [35], and NeuroMorph [36].

To evaluate all correspondences, we adopt the Princeton protocol, a method widely used in recent works [24, 25, 27, 29, 35]. Specifically, we calculate the pointwise geodesic error between the predicted maps and the ground truth map, normalizing it by the square root of the target shape's total surface area.

Results are presented in Table 1. As can be seen, our method outperforms axiomatic approaches and is comparable or superior to some training-based methods, particularly on the FAUST and SCAPE datasets. When it comes to the SHREC dataset, our method achieves state-of-the-art results across *all methods*. Most learning-based methods stumble over this challenging dataset, even though they have been trained on many training examples. This is because of the wide mix of poses and identities in the

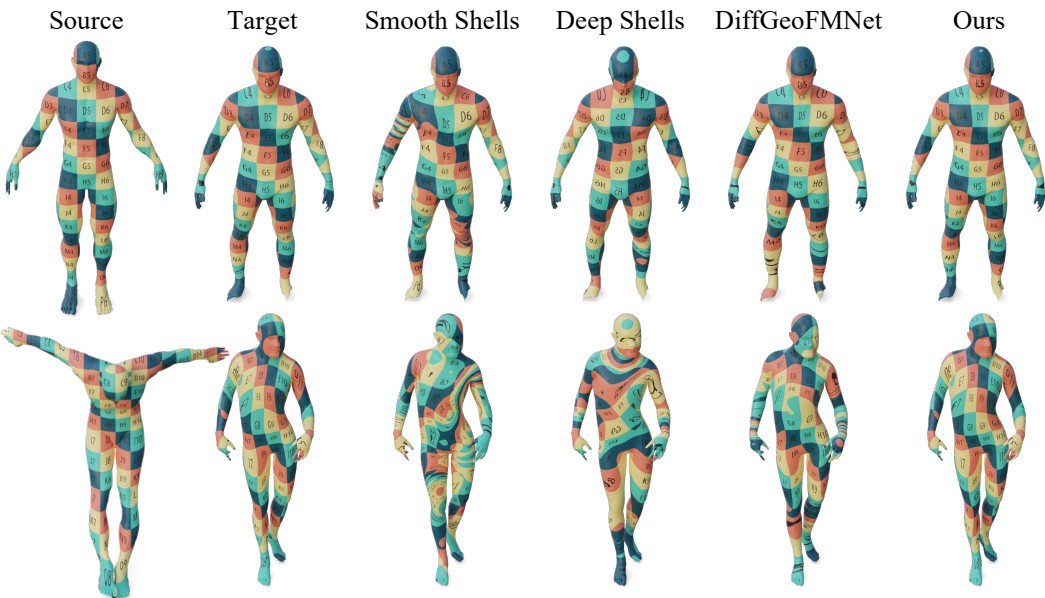

| Source | Target | Smooth Shells | Deep Shells | DiffGeoFMNet | Ours |

Figure 4: **Qualitative Results** from the **SHREC** dataset. The correspondences are visualized by transferring a texture through the map. Our method results in visually superior outcomes.

Table 2: **Quantitative results on the non-isometric benchmark**. We highlight the **best** and the second best results. Our method achieves competitive results with trained methods.

|  | METHOD | SMAL |
|---|---|---|
| *Sup* | FMNet [24] | 47.1 |
|  | GeoFMNet [25] | 8.8 |
|  | DiffGeoFMNet [64] | **6.6** |
|  | LIE [82] | 20 |
| *Unup* | WSupFMNet [29] | 13.3 |
|  | Deep Shells [35] | 15.2 |
|  | NeuroMorph [36] | 23.1 |
| *Axiom* | Smooth Shells [55] | 16.3 |
|  | SNK (Ours) | **9.1** |

dataset. This highlights how our approach, which focuses on overfitting a single pair, is particularly effective in such challenging cases.

In Fig. 4, we showcase point-to-point maps generated by our method on the **SHREC** dataset. We compare these maps to those produced by the best-performing baseline method across each category - axiomatic, unsupervised, and supervised. Upon inspection, it is apparent that our method attains highly accurate correspondences. In contrast, the baseline methods either fail to achieve a precise matching or suffer from symmetry flip issues.

## 5.3 Non-Isometric Shape Matching

In the non-isometric, non-rigid context, we adhered to the same evaluation protocol and present our findings in Table 2. It can be seen from these results that our method outperforms both the axiomatic and unsupervised approaches, and competes strongly with, or even surpasses, some supervised techniques. This further demonstrates the efficacy and versatility of our method across diverse shape-matching scenarios.

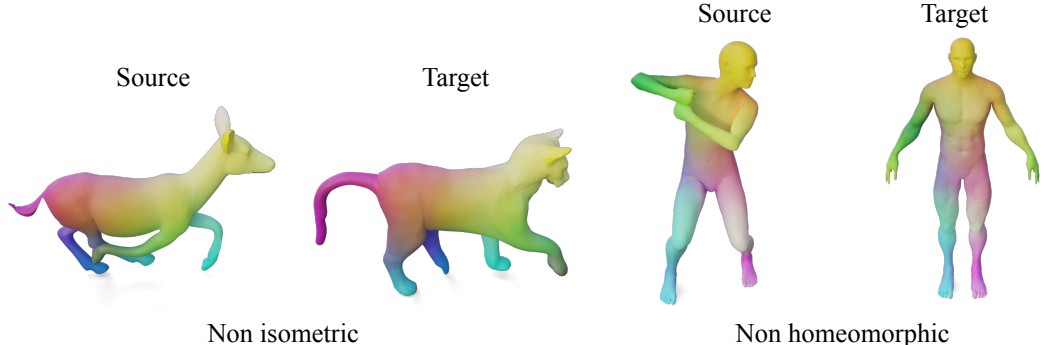

Figure 5: Examples of non-isometric and non-homeomorphic matching on the **SMAL** and **SHREC** datasets, respectively.

In Fig. 5, we present qualitative examples of non-isometric and non-homeomorphic matching from the **SMAL** and **SHREC** datasets, respectively. The results further highlight the resilience of our approach to various deformations and topological changes.

### 5.4 Ablation Studies & Additional Details

In the supplementary material, we present an ablation study that underscores the efficacy of each component introduced in our methodology. Additionally, we provide comprehensive details regarding the implementation process, the computational specifications involved, as well as some qualitative results.

**Importance of the neural prior** In the supplementary we also evaluate the importance of using neural networks for training our method. In particular, we show that when unconstrained optimization replaces neural networks, the results degrade significantly. This illustrates the importance of the "neural prior" [83, 84] employed by our method and further highlights the difference between our approach and purely axiomatic methods.

## 6 Conclusion & Limitations

In this work, we introduced **SNK**, a novel approach to zero-shot shape matching that operates independently of prior training. Our method hinges on the precise alignment of a pair of shapes, enabling the source shape to be deformed to match the target shape without any user intervention. By integrating both spatial and spectral losses, and architecting a new decoder that encourages smooth and visually pleasing deformations, our method has proven its effectiveness. Extensive experiments reveal that our approach competes with, and often surpasses, many supervised techniques, even achieving state-of-the-art results on the **SHREC** dataset.

Despite its success, **SNK** has certain limitations we aim to address in future research. Firstly, our method is currently tailored towards complete shape correspondence, necessitating adaptation for partial shape-matching scenarios. Secondly, while our method's convergence time of approximately 70 seconds per shape outpaces many competitors, exploring ways to further expedite this process would be beneficial. Finally, our method utilizes DiffusionNet as a backbone, which theoretically could accommodate both point clouds and triangular 3D meshes. Consequently, extending our approach to encompass other types of representations could prove useful.

**Acknowledgements** The authors would like to thank the anonymous reviewers for their valuable suggestions. Parts of this work were supported by the ERC Starting Grant No. 758800 (EXPROTEA) and the ANR AI Chair AIGRETTE.

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

# Supplementary Materials for: Shape Non-rigid Kinematics (SNK)

**Souhaib Attaiki**
LIX, École Polytechnique, IP Paris
attaiki@lix.polytechnique.fr

**Maks Ovsjanikov**
LIX, École Polytechnique, IP Paris
maks@lix.polytechnique.fr

In this document, we have gathered all the results and discussions that, due to page limitations, were not included in the main manuscript.

More precisely, we initially detail the specifics of our implementation in Appendix A. Following this, we present a comprehensive analysis of the components introduced in our methodology in Appendix B. Further examination of our architecture's components can be found in Appendix C. Lastly, we include some qualitative results in Appendix D.

## A  Implementation Details

### A.1  Network Architecture

In Section 4 of the main paper, we elaborated on our architecture, which is comprised of two primary modules: the functional map module and the shape reconstruction module.

For the functional map module, we utilized DiffusionNet [64] as a feature extractor, using the authors' original implementation [1]. It consists of four blocks, each with a width of 128, and the feature dimension $F_i$ is likewise 128. The functional map block possesses a dimension of $k = 30$, and we used $\lambda = 10^{-3}$ in Equation 1, and $\tau = 0.07$ in Equation 4 from the main paper.

The shape reconstruction module employs an encoder via DiffusionNet, comprising four blocks and possessing a width of 128, with an output dimension $d_1 = 512$. We derive the latent code $l$ by performing max-pooling over the vertex dimension.

In terms of PriMo modeling, we construct a volumetric layer by (virtually) extruding the input mesh along the vertex normals. This process generates rigid prisms for each mesh face, with each prism having a height, $h$, set to 0.02 (see Figure 2 of the main text).

Our novel decoder, the prism decoder, relies on DiffusionNet at its core. Given the source shape and the latent code $l$, we duplicate the latter, concatenate it with the XYZ coordinates of the source shape, and then process them with a DiffusionNet composed of four blocks with a width of 128. The output is a pointwise feature $D \in \mathbb{R}^{n_2 \times d_2}$ with $d_2 = 512$. We transform these features into face-wise features $G$, as described in the main paper. These features are subsequently refined with a pointwise MLP, which consists of four layers of Linear layer and ReLU layer; however, the last layer does not contain a ReLU. The width of the MLP gradually narrows from 512 to 12 in the final layer. We obtain the matrix $R$ in Equation 5 from the main paper by initially performing an SVD decomposition [75] of $R_0$ such that $R_0 = \mathbf{U}\mathbf{D}\mathbf{V}^\top$. The solution is then obtained by $R = \mathbf{U}\mathbf{S}\mathbf{V}^\top$ with $\mathbf{S} = diag(1, 1, det(\mathbf{U})det(\mathbf{V}))$. Employing the predicted translation and rotations matrices $t_i$ and $R_i$ for each face (considered a prism in the PriMo design), we rigidly deform the source mesh faces. The deformed surface mesh is finally derived from the resulting prisms by updating the position of each vertex using the average transformation of its incident prisms.

For all our experiments, we utilized the Adam optimizer [85], with a learning rate preset to 0.001. With respect to the losses in Equation 6 from the main text, we used $\lambda_{\mathrm{mse}} = \lambda_{\mathrm{fmap}} = \lambda_{\mathrm{cycle}} = \lambda_{\mathrm{primo}} = 1$.

---

[1] https://github.com/nmwsharp/diffusion-net

Table 3: **Runtime comparison**. A comparison of the runtime of various baseline methods with our method on the **FAUST** dataset. Time is presented in seconds.

| METHOD | Runtime (s) |
|---|---|
| BCICP [19] | 780 |
| Smooth Shells [55] | 430 |
| **SNK** (Ours) | 70 |

For every new pair, we trained our network over the course of 1000 iterations, terminating the training if no improvement in the loss was observed for over 100 consecutive iterations.

### A.2 PriMo Loss Implementation

In the main paper, Equations 2 and 3 introduced the formulation for the PriMo energy. In this section, we further detail the practical computation of this energy.

Given two functions $a(\mathbf{u})$ and $b(\mathbf{u})$ defined by bi-linear interpolation of four values $\{a_{00}, a_{10}, a_{01}, a_{11}\}$ and $\{b_{00}, b_{10}, b_{01}, b_{11}\}$ respectively, we define the scalar product [86]:

$$
\begin{aligned}
\langle a, b \rangle_2 &:= \int_{[0,1]^2} a(\mathbf{u}) \cdot b(\mathbf{u}) d\mathbf{u} \\
&= \frac{1}{9} \sum_{i,j,k,l=0}^{1} a_{ij} \cdot b_{kl} \cdot 2^{(-|i-k|-|j-l|)}
\end{aligned}
\tag{7}
$$

Next, referring to Figure 2 in the main paper, the energy $E_{ij}$ is simply defined as:

$$
E_{ij} = \langle \mathbf{f}^{i \to j} - \mathbf{f}^{j \to i}, \mathbf{f}^{i \to j} - \mathbf{f}^{j \to i} \rangle_2
\tag{8}
$$

Here, $\mathbf{f}^{i \to j} = \{\mathbf{f}_{00}^{i \to j}, \mathbf{f}_{10}^{i \to j}, \mathbf{f}_{01}^{i \to j}, \mathbf{f}_{11}^{i \to j}\}$ represents the four corners of the face of prism $P_i$ that is adjacent to prism $P_j$.

### A.3 Computational Specifications

We carry out all our experiments using Pytorch [87] on a 64-bit machine equipped with an Intel(R) Xeon(R) CPU E5-2630 v4 operating at 2.20GHz and an RTX 2080 Ti Graphics Card. For comparison, we use the original code provided by the authors of each competing method and adhere to the optimal parameters reported in their respective papers.

### A.4 Runtime Comparison

Our method's runtime performance, compared to various baselines, is analyzed in Tab. 3. Specifically, we assess the overall runtime for all the baselines on the **FAUST** dataset, which consists of shapes averaging around 5000 vertices. The average time per pair, reported in seconds, demonstrates that our method outpaces the competition in terms of speed.

## B Ablation Study

In Section 4 of the main text, we introduced the components of our methodology. In this section, we investigate the impact of each component independently. For illustrative purposes, we utilize the near-isometric matching experiment on the **FAUST** dataset (as referenced in Section 5.2 of the main text). Nevertheless, analogous conclusions can be drawn in other scenarios.

Initially, we examine the novel decoder architecture we proposed. In this case, we maintain all other aspects of our method and solely modify the decoder. We experiment with a decoder based on a

Table 4: **Ablation Study**. This table displays an ablation study of the various components of our method, conducted on the FAUST dataset. The study shows that every component is crucial to the overall performance. The **best** result is highlighted.

| | METHOD | FAUST |
|---|---|---|
| *Decoder* | ■ MLP | 35.9 |
| | ♦ DiffusionNet | 13.4 |
| | ⋆ DiffusionNet + ARAP | 14.8 |
| *Encoder* | ⬠ Latent Code | 4.0 |
| *Losses* | ● Chamfer loss | 80.7 |
| | ⊕ Spectral only | 8.7 |
| | ▲ Ours (without $\mathcal{L}_{\mathrm{primo}}$) | 6.5 |
| | ♥ Ours (without $\mathcal{L}_{\mathrm{cycle}}$) | 2.7 |
| | **SNK (Ours)** | **1.9** |

Multilayer Perceptron (MLP) network (■), and one based on the DiffusionNet network (♦), both possessing a comparable number of parameters to our decoder. Furthermore, given that several methods use the As-Smooth-As-Possible (ARAP) [65, 35, 36] energy for deformation regularization, we also compare our approach with a DiffusionNet decoder supplemented with ARAP loss (⋆).

Concerning the encoder, besides the shape encoder design we employed, we also explored the option of directly learning the latent code (marked with ⬠) in a way similar to DeepSDF [88].

With respect to the loss functions, we compare our method against an approach that exclusively overfits the spectral losses [28], and then report the result of converting the predicted functional maps into point-to-point (⊕). Additionally, we also report the outcome of training our network with the unsupervised Chamfer loss as a substitute for the Mean Squared Error (MSE) loss $\mathcal{L}_{\mathrm{mse}}$ that we used (●). Moreover, we experiment with our network trained without the PriMo loss $\mathcal{L}_{\mathrm{primo}}$ (▲) and finally without the cycle loss $\mathcal{L}_{\mathrm{cycle}}$ (♥).

The results are summarized in Tab. 4, illustrating that all the components we incorporated are crucial for optimal performance. Specifically, the choice of the decoder is vital. Both the MLP and DiffusionNet decoders failed to match the results of our decoder, even when all our loss functions were used. The ARAP loss, in particular, hindered the performance of the DiffusionNet decoder, a contrast to its typical application in network training on a dataset.

Regarding the encoder, the shape encoder design outperforms the latent code design. This superior performance can be attributed to the shape encoder's ability to utilize the input shape as an added information source. This advantage is further enhanced by the intrinsic regularization provided by the network architecture, often referred to as the neural bias [83, 84].

In terms of the loss functions, training our architecture with the Chamfer loss was completely unsuccessful as it does not preserve the order of the points and is therefore unsuitable for matching. Using only the spectral loss yields decent results but falls short of state-of-the-art performance. This illustrates the importance of the interplay between spectral and spatial losses for optimal performance. The PriMo Energy plays a key role as, without it, the decoder fails to produce smooth, natural-looking deformations [67] that facilitate learning. Finally, the cycle loss proves critical in refining the point-to-point maps and achieving state-of-the-art results.

## C  In-Depth Exploration of SNK's Components

### C.1  The Feature Extractor

The feature extractor plays a pivotal role in our architecture. The features it extracts are crucial for predicting both the fmap and the p2p maps, which in turn influence the loss. Consequently, consistency between the features of the two shapes is essential. To elucidate this, we have provided a

Features 1      Features 2

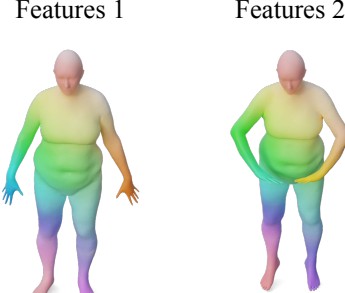

Figure 6: Consistency between extracted features for source and target shapes.

visualization in Fig. 6 that highlights the congruence of features produced by the feature extractor for a shape pair. In this representation, we reduced the feature dimensions using PCA and then portrayed them as RGB values. The visualization readily demonstrates that the features from both shapes align closely, highlighting congruent areas.

## C.2 SNK's Priors

In this section, we detail the priors employed by our method and highlight the unique neural prior that sets our approach apart from axiomatic methods.

- Elastic Energy as a Prior: PRIMO being an elastic energy, it penalizes non-linear stretching and bending. This directs our optimization towards smooth and realistic deformations.
- Cyclic Bijectivity via the cycle consistency loss.
- Near-isometry: Our method weakly enforces near-isometry in fmap predictions through Laplacian commutativity in a reduced ($k = 30$) functional basis. It should be noted that using elastic energy and weakly promoting isometries with low-frequency functional maps for non-isometric matching has precedents in works like [89, 90, 91, 55].
- Neural Networks: Finally, a subtle yet essential prior in our method is in our use of a neural network for feature extraction. Our approach follows studies like Deep Image Prior [84] and Neural Correspondence Prior [83] which show that neural networks can act as strong regularizers, generating features that enable plausible mappings even without pre-training

To demonstrate the neural prior empirically, we conducted two additional experiments:

- HKS-based Features: We replaced our feature extractor with the more traditional HKS features.
- Free Variables: We treat $F_1$ and $F_2$ as free variables, optimized via gradient descent without using a neural network.

Our findings (see Tab. 5) show that without the neural network regularization, the free variable model fails to converge. The HKS-based approach performs, but significantly worse than ours. This suggests neural networks offer extra regularization, surpassing traditional handcrafted features.

Overall, our method's priors are synergistic and versatile, making our approach applicable to a broad range of shapes.

## D Qualitative Results

In this section, we present a selection of visual results derived from our proposed method.

Firstly, in Fig. 7, we present the reconstructed shapes from the **SCAPE** dataset as produced by our method when overfitting a shape pair. This reconstruction is subsequently used to perform matching

Table 5: Ablation on the neural prior.

| METHOD | FAUST |
|---|---|
| HKS features | 20.3 |
| Free Variables for Features | 64.0 |
| **SNK** (Ours) | **1.9** |

Target        Source        Reconstruction

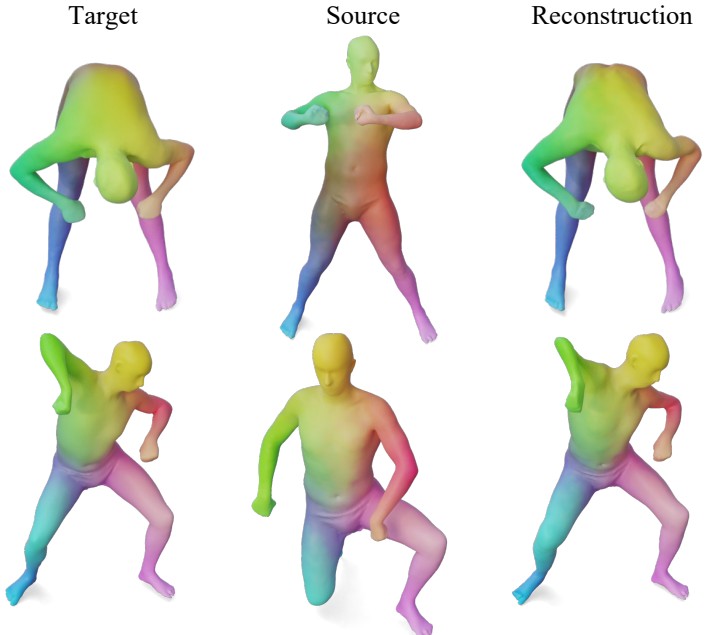

Figure 7: **Shape Reconstruction** from the **SCAPE** dataset: The figure showcases the results of deforming the source shape to match the target shape using our architecture. It's noteworthy how well the shape has been reconstructed, a process that was completed in an entirely unsupervised manner.

using the nearest-neighbor approach. The high quality of the reconstruction is evident, even when dealing with challenging input pairs where the source and target pose exhibit significant variance.

Secondly, we inspect the quality of the shapes reconstructed by our novel decoder architecture during training. For this experiment, we deform a source shape to match a target shape using only the $\mathcal{L}_{\mathrm{mse}}$ loss. Instead of utilizing the map generated by the functional map module, we employ the ground truth map, visualizing various shapes produced during the **optimization process**. We compare our decoder to a DiffusionNet-based decoder and a decoder utilizing DiffusionNet with ARAP loss applied, aligning with the approach in Appendix B. These results are illustrated in Fig. 8 using a shape pair from the **FAUST** dataset. Our method is the only one to produce natural-looking, smooth deformations, which in turn results in superior matching outcomes.

Lastly, we showcase some failure cases of our method on the partial setting. In fact, our method is geared toward complete shape matching, but in Fig. 9, we showcase our method's performance on three partial pairs from the Shrec 16' Cuts and Holes subsets [49]. Our method performs satisfactorily when dealing with moderate partiality.

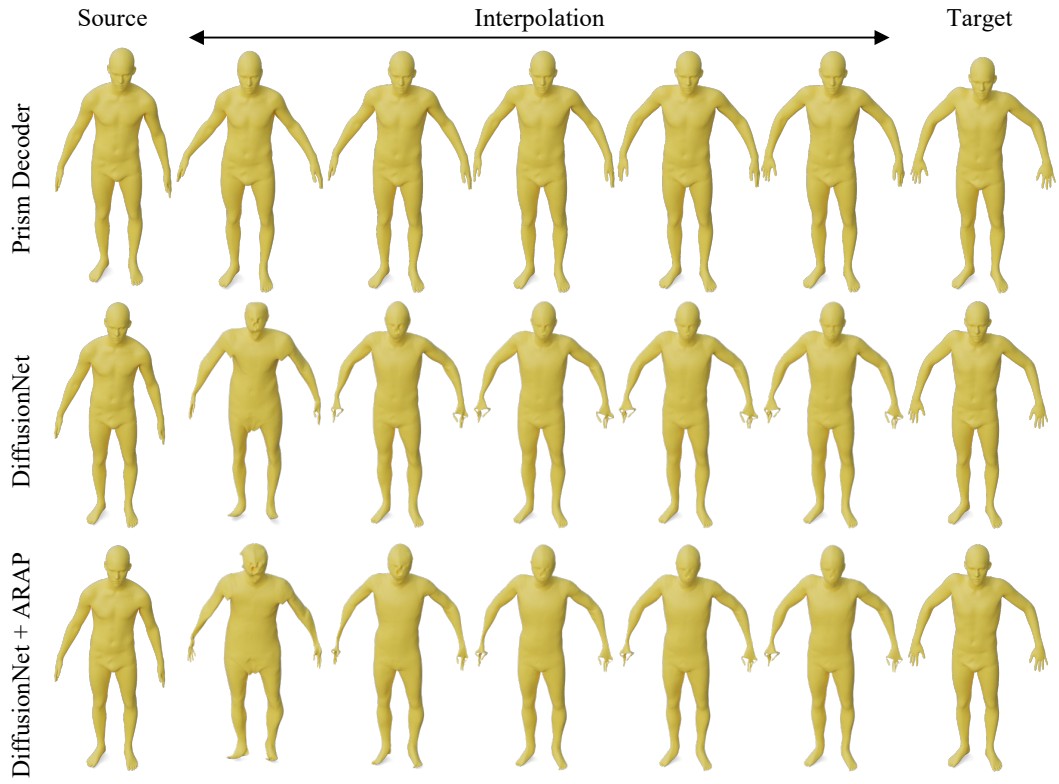

Figure 8: **Shape Interpolation** using the **FAUST** dataset: Various decoders are employed to interpolate between a source and target shape. Compared to the baselines, our method yields smooth and natural-looking deformations.

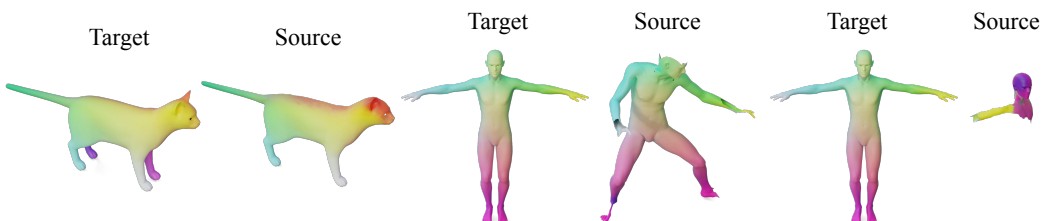

Figure 9: Our method is tailored toward complete shape matching. Here we showcase the results of our method on some partial pairs from the Shrec 16 partial benchmark.

