# OpenReview forum: "Shape Non-rigid Kinematics (SNK): A Zero-Shot Method for Non-Rigid Shape Matching via Unsupervised Functional Map Regularized Reconstruction"
_NeurIPS.cc/2023/Conference — NeurIPS 2023 poster_

### Official Review · Reviewer_nxSf · 2023-06-28

**Soundness:** 4 excellent
**Presentation:** 4 excellent
**Contribution:** 3 good
**Rating:** 6
**Confidence:** 5

**Summary:**

The paper proposes a zero-shot method for non-rigid shape matching based on
unsupervised functional map regularisation and non-rigid shape deformation regularisation.
The proposed method combines both intrinsic information from the functional map regularisation and
the extrinsic information from shape deformation regularisation (i.e. PriMo energy) leading to a better matching performance compared to methods solely based on either intrinsic or extrinsic information from the 3D shapes.
In the standard shape matching benchmarks, the proposed method demonstrates better matching performance
compared to axiomatic approaches and competitive results in comparison to both supervised and unsupervised learning-based methods.


**Strengths:**

1. The paper is well-written and easy to follow. Both the motivation and the proposed method are well explained in the main paper. The technical details are also well illustrated in the supplementary material.
2. Unlike most existing shape matching methods solely based on functional map regularisation, the proposed method also considers spatial alignment and adds the shape deformation regularisation to incorporate also the extrinsic/spatial information leading to a better matching performance.
3. Instead of using common deformation regularisation such as ARAP, Laplacian, the paper utilises the PriMo energy leading to a more smooth and natural shape deformation, as demonstrated in the supplementary material.

**Weaknesses:**

1. The paper claims that the method is a zero-shot method for non-rigid shape matching. But the main difference between
the proposed method and the previous unsupervised methods is not well discussed. In theory, all previous unsupervised
methods can also be used as a zero-shot method to optimise only for a single shape pair. More explanation and experiments should be provided.
2. In the supplementary material, it shows the average runtime on the FAUST dataset is about 70s, which is much slower compared to existing learning-based methods. Prior unsupervised works (e.g. Cao et al. Unsupervised Deep Multi-Shape Matching, ECCV 2022) typically train on the training data and utilise test-time-adaptation for a few iterations to reduce runtime. More discussion and experiments about the runtime is expected to be provided.
3. Unlike purely intrinsic/spectral methods, the proposed method also aims to spatially align two shapes.
Therefore, more insights about the rotation robustness of the proposed method should be given.

**Questions:**

1. Even though in the limitation part, the paper claims that the proposed method is specifically tailored for complete shape matching, it is also recommended to include some (potential failure) matching results of partial shapes in the supplementary material. This will benefit follow-up works addressing partial matching.
2. What is the number of the LBO eigenfunctions for functional map computation? Prior methods like Smooth Shells/Deep Shells optimise the matching with gradually increasing number of the LBO eigenfunctions - is the proposed method also robust to the choice of the number of LBO eigenfunctions?

**Limitations:**

1. The proposed method is specifically tailored for complete shape matching, so more adaptations should be taken for partial shape matching.
2. The runtime of the proposed method takes 70s for shapes with ~5000 vertices, which is much slower compared to existing learning-based methods.
3. The proposed method is tailored to shapes represented as triangle mesh, so the extension to other data representation like point cloud could be the future work.

---

> ### Author Rebuttal · Authors · 2023-08-09
>
> We thank the reviewer for their detailed and thoughtful comments and suggestions. We invite the reviewer to consider the general feedback that we provided to all reviewers in the "Author Rebuttal" answer. Below are our answers to the reviewers’ questions.
>
>
> **Q1: Difference between the proposed method and other unsupervised methods**
>
> Thank you for pointing this out. In the introduction (L29-33), we highlighted that while unsupervised techniques can work without labeled data, their best performance often requires training on numerous shape examples for extended periods [35, 36]. Such methods, once trained, can exhibit biases towards their training datasets, as seen with DeepShells' strong results on Faust and Scape but poorer performance on Shrec. Unlike them, our zero-shot approach requires no prior extensive training, allowing immediate use on new shape pairs and avoiding dataset-specific biases.
>
> On using unsupervised methods as zero-shot techniques, we agree. Our ablation study in the supplementary material's Table 4 (specifically row 5) shows that while possible (training an unsupervised approach individually on each shape pair), these techniques don't rival the efficacy of our method, underlining our approach's uniqueness and benefits.
>
>
> **Q2: Runtime Comparison Between the Proposed Method and Some Unsupervised Methods**
>
> We appreciate your observation. While our method may seem slower at test time compared to some unsupervised techniques, a direct comparison might be skewed. Our method is zero-shot, meaning it handles new shape pairs without prior training. A more apt comparison might be with axiomatic methods, which also work without training. This is the comparison we have shown in Table 3 of the supplementary material.
>
> On the other hand, unsupervised methods require significant datasets and long training times for optimal performance. Comparing them to our approach without accounting for their training context might not give a full perspective.
>
> **Q3: Rotation robustness of the proposed method**
>
> Thank you for highlighting this aspect. As referenced in L262, both our approach and unsupervised methods utilize datasets aligned in accordance with previous studies [29,35,62,36,53]. To delve deeper into rotation robustness, we conducted an experiment. We randomly selected 10 shape pairs from the Faust dataset and computed the point-to-point maps with our method. Concurrently, we progressively rotated one shape around the up-axis from 0 to 45°. The outcomes, depicted in Figure 9 of the "Author Rebuttal" answer, indicate a slight performance drop as rotation increases. It's pivotal to note that combining intrinsic and extrinsic properties is what empowers our approach to deliver robust results. We'll clarify this further in our revised manuscript.
>
> **Q4: Matching results on partial shapes**
>
> Thank you for your input. In Figure 10 of the "Author Rebuttal" answer, we showcase our method's performance on three partial pairs from the Shrec 16' Cuts and Holes subsets. Our method performs satisfactorily when dealing with moderate partiality.
>
>
> **Q5: Robustness to the number of eigenfunctions**
>
> Thanks for highlighting this. As noted in the supplementary material (L12), we use 30 eigenfunctions, aligning with many recent studies using deep functional maps, such as [25,27,26,43,29,62,75]. We also tested with 50 eigenfunctions, yielding comparable results (1.9 geodesic error on the Faust dataset). This will be clarified in the revised version.
>
>
> **Q6: The proposed method is tailored towards complete shapes**
>
> We acknowledge your point. While our method is primarily built for complete shapes, adapting it for partial shape-matching is a viable next step. We consider employing strategies like in DPFM [27], by refining functional map estimation for partial contexts and predicting a pointwise mask to highlight aligning regions. This mask could then refine the MSE loss. However, such adaptations are non-trivial and present an exciting direction for future work.
>
>
>
> **Q7: The proposed method is tailored towards triangular meshes**
>
> Thank you for noting this. Our method is primarily designed for triangular meshes. Adapting it to point clouds would entail using feature extractors like DGCNN [85] or PointMLP [86] tailored for such data. We see this as a promising direction for future exploration and plan to delve into it in subsequent research.

---

> > ### Comment · Reviewer_nxSf · 2023-08-15
> >
> > Thanks for the clear explanation of my questions/concerns.
> > I acknowledge that the proposed method achieves better zero-shot matching performance by combining both spectral and spatial alignment.
> > However, I still would like to know if the proposed method is trained in a standard unsupervised manner instead of zero-shot, how well would it perform, especially for the test data.

---

> > > ### Author Response · Authors · 2023-08-20
> > > **Clarifications on Zero-Shot Versus Standard Unsupervised Training Performance**
> > >
> > > Thank you for the insightful question. To address your query regarding our method's performance under standard unsupervised training, we offer the following clarifications:
> > >
> > > Firstly, it's important to emphasize that our primary focus in this paper has been the zero-shot setting. Our approach is particularly geared towards scenarios in which there are no extensive datasets or computational resources available for training.
> > >
> > > Moreover, as illustrated in our paper, methods that rely on training can sometimes become biased toward the training data. This can lead to potential failures when encountering new data, as exemplified by the performance of models like DeepShells. In contrast, our SNK method evaluates each shape pair individually, making it unaffected by prior data or remeshing. Remarkably, even under these constraints, SNK demonstrates competitive results, matching or even surpassing methods that have undergone extensive training, achieving state-of-the-art results on the SHREC'19 dataset (please see Table 1 in the main paper.).
> > >
> > > Nevertheless, following your suggestion, we did run an experiment on the FAUST dataset for additional comparisons. In this test, we trained our method using the training subset of FAUST using our unsupervised losses and then tested on its corresponding test set, as done in previous works in this domain. Preliminary results showed a score of 1.6 for the trained method, while our zero-shot approach recorded a score of 1.9. We'd like to point out that these experiments were conducted under limited time constraints, so we did not embark on any extensive optimization or hyperparameter tuning. Therefore, there's potential for further enhancement of these results.
> > >
> > > In conclusion, while these additional results provide interesting insights, they do not shift the primary contribution of our work, which remains the zero-shot approach. We trust that this response addresses your question adequately, and we remain available for any further discussions.

---

### Official Review · Reviewer_j61z · 2023-07-04

**Soundness:** 3 good
**Presentation:** 4 excellent
**Contribution:** 2 fair
**Rating:** 5
**Confidence:** 5

**Summary:**

A zero shot method for computing correspondences between meshes is proposed. The core idea is to use DiffusionNet to produce features which are then used to produce a functional map from which a p2p mapping is produced. Using it, a deformation module using Primo is employed to deform the meshes to one another. This deformed shape is then used to reevaluate the functional maps and the process continues, training an autoencoder to predict the features that drive the correspondence and deformation.

**Strengths:**

Zero-shot correspondence computation is an important problem to solve in cases where no object categories are known. The combination of an extrinsic deformation module with an intrinsic functional maps approach is appealing and novel as far as I can tell. The paper is written in a clear manner for the most part (see below).

**Weaknesses:**

I have several concerns:
* In this context, "zero shot" essentially means "no use of semantic information" (as opposed to, e.g., zero shot stemming from using some existing module that has semantic knowledge and applying it to the correspondence problem without further training). Thus, the term used in the paper for "axiomatic" methods simply means ones using geometric priors rather than semantic ones. As such, I am uncertain why use the word "zero shot" at all -- this is simply an optimization algorithm for shape correspondence. As such it lies within a different research area than learning which encompasses many other works. Two recent ones are ENIGMA and Neural Surface Maps. The advantage of this approach of theirs is not immediately apparent and they should be discussed at least, if not compared to.
* It is unclear to me what is the exact prior being enforced here. At first I thought it's a an isometric-distortion minimization prior but the authors show results on non-isometric datasets, so it is unclear to me what is the prior here and what is the justification for it. Without a justified prior nor semantics, it seems concerning the method solely relies on empirical success. It is further unclear what specific class of deformations is Primo regularizing for. It does seem to be for reducing isometric distortion here, thus I do not fully understand the claim to work on a non-isometric case. I suspect that the method is still working on rather-isometric cases - SMAL's 4-legged animals are still rather isometric. Again in this context, ENIGMA shows success and seems like a good comparison.
* Additionally, the method is currently limited to matching meshes that are homeomorphic and are made up of one connected component - this is not mentioned in the paper and is somewhat limiting.
* in terms of novelty, the method is simply a mix of several techniques - DiffusionNet, Functional Maps, Primo, which indeed work well together, but there does not seem to be a great novel insight beyond combining them to have a method to predict correspondences and then update a deformation prediction, which is a very classic idea (in essence, a non-rigid ICP approach where the closest point is replaced with a functional map prediction).

**Questions:**

how is the use of the low eigenspectrum of the laplacian justified without assuming near-isometry?
why is the autoencoder needed? why couldn't the functional maps and deformation module be optimized in block-descent manner directly?

**Limitations:**

yes

---

> ### Author Rebuttal · Authors · 2023-08-09
>
> We thank the reviewer for their detailed and thoughtful comments and suggestions. We invite the reviewer to consider the general feedback that we provided to all reviewers in the "Author Rebuttal" answer. Below are our answers to the reviewer's questions.
>
> **Q1: The Use of "Zero-Shot"**
>
> Thank you for your remark on our terminology. We use "Zero-Shot" for two main reasons: (1) Instead of relying on fixed input features like HKS or SHOT, we use a deep neural network for feature computation, and (2) Our use of a neural network provides an important prior for our method, compared to purely geometric `axiomatic methods` (please see also our response to Q3 below). Nevertheless, we recognize the potential ambiguity and will articulate this more clearly in our revised manuscript.
>
> **Q2: ENIGMA & NSM**
>
> Thank you for referencing ENIGMA and Neural Surface Maps (NSM). In our comparison, we focused on the most recent SOTA techniques that evaluate their results on standard dense non-rigid benchmarks, such as FAUST, SCAPE, SHREC’19, etc. Nevertheless, we agree that these are relevant prior methods and will include a discussion in the final version. Briefly,
>
> 1. ENIGMA: While ENIGMA is indeed a relevant baseline, its code isn't publicly accessible, making direct comparison non-trivial. We've requested the authors for the implementation and will update accordingly.
>
> 2. NSM: NSM's main objective is 3D mesh representation rather than shape matching. When applied to shape-matching, NSM necessitates input keypoints, making it semi-supervised. Furthermore, NSM's cited correspondence computation between a shape pair ranges from 6-10 hours.
>
> **Q3: Underlying Priors**
>
> We're grateful for the insightful queries. Here's a concise overview of our method's key priors:
>
> 1. Elastic Energy as a Prior: PRIMO being an elastic energy, it penalizes non-linear stretching and bending. This directs our optimization towards smooth and realistic deformations.
>
> 2. Cyclic Bijectivity via the cycle consistency loss.
>
> 3. Near-isometry: Our method *weakly* enforces near-isometry in fmap predictions through Laplacian commutativity in a *reduced* (k=30) functional basis.  It should be noted that using elastic energy and weakly promoting isometries with low-frequency functional maps for non-isometric matching has precedents in works like [81,82,53] and ENIGMA.
>
> 4. Neural Networks: Finally, a subtle yet *essential* prior in our method is in our use of a neural network for feature extraction. Our approach follows studies like Deep Image Prior [83] and Neural Correspondence Prior [84] which show that neural networks can act as strong regularizers, generating features that enable plausible mappings even without pre-training.
>
> To demonstrate this empirically, we conducted two additional experiments:
>
> a. HKS-based Features: We replaced our feature extractor with the more traditional HKS features.
>
> b. Free Variables: We treat F1 and F2 as free variables, optimized via gradient descent *without using a neural network*.
>
> Our findings (see Table 5 in the "Author Rebuttal") show that without the neural network regularization, the free variable model fails to converge. The HKS-based approach performs, but significantly worse than ours. This suggests neural networks offer extra regularization, surpassing traditional handcrafted features.
>
> Overall, our method's priors are synergistic and versatile, making our approach applicable to a broad range of shapes. We'll highlight this analysis in our revised manuscript.
>
> **Q4: Novelty Concerns**
>
> Thank you for feedback on our method. While building on prior work, our contributions are distinct:
>
> 1. Primo Energy Adaptation: We've reimagined Primo Energy's usage. Contrary to the original paper's user-constrained approach, ours is fully automatic, in a *deep learning setting*, marking a departure from its initial intent. Moreover, our deformation model uses a face-based representation, with strong regularization, which distinguishes our method from most previous works in this domain that use direct vertex-based deformations.
>
> 2. Novel approach and loss integration: (a) Our use of a neural network for feature extraction in the Zero Shot setting, as well as (b) our combination of extrinsic and intrinsic losses in a deep unsupervised setting, are both novel. Furthermore, we believe that demonstrating that the resulting approach can perform on par or even better than data-driven methods, but in a zero-shot setting, constitutes an important contribution to the domain.
>
> In essence, our novelty stems from the innovative integration and adaptation of existing components, bringing significant progress to shape matching.
>
> **Q5: Isometry, Topology, & Multiple Components**
>
>  * Isometry: The SMAL dataset is diverse, including animals like lions, cats, and deer in different poses. Our method's successful matches here demonstrate its robustness to non-isometric challenges.
>
>  * Homeomorphism: Our approach is not restricted to homeomorphic shapes. The SHREC dataset, which tests among other *topological* changes, proves our method's adaptability. Please see Figure 8 for visual examples of our method from SMAL and SHREC.
>
>  * Multiple Components: We recognize DiffusionNet's limitation here. Considering alternative feature extractors or viewing shapes as point clouds might help. However, this challenge isn't unique to us; other recent SOTA methods using DiffusionNet [27,83,25,62,26] share it.
>
> **Q6: Sequential Optimization**
>
> The deformation module relies on the fmap module's outputs, so training it first isn't viable. Our ablation study (Table 4) shows an 8.7 error rate on Faust when only training the fmap, versus 1.9 with our full method. This quality wouldn't effectively train the deformation module, evidenced by a 9.2 error when we tried. The synergy between intrinsic and extrinsic modules is vital for optimal results.
>
> **Q7: Autoencoder Necessity**
>
> Due to length limits, refer to our response to Q2 from reviewer **osbV**.

---

### Official Review · Reviewer_E6YV · 2023-07-04

**Soundness:** 3 good
**Presentation:** 2 fair
**Contribution:** 3 good
**Rating:** 5
**Confidence:** 4

**Summary:**

The paper proposes a novel zero-shot method for non-rigid shape matching that eliminates the need for extensive training or ground truth data. The proposed method, Shape Non-rigid Kinematics (SNK), operates on a single pair of shapes and employs a reconstruction-based strategy using an encoder-decoder architecture, which deforms the source shape to match the target shape closely. SNK demonstrates competitive results on traditional benchmarks, simplifying the shape-matching process without compromising accuracy.


**Strengths:**

The paper introduces a novel method for non-rigid shape matching that does not require extensive training data, making it more practical for many applications. The proposed method also combines the benefits of axiomatic and learning-based approaches and addresses their limitations. Moreover, the paper introduces a new decoder architecture that facilitates the generation of smooth and natural-looking deformations. The proposed method is thoroughly evaluated and achieves state-of-the-art results on the SHREC dataset.


**Weaknesses:**

While the paper highlights the strengths of the proposed method, it lacks a thorough comparison with closely related work. For instance, although several learning-based methods are presented, there is no detailed comparison, such as a visual comparison and evaluation of other datasets (recommend moving some results to the main paper). Also, the limitations of the proposed method are mentioned but need to be adequately discussed.

For performance, the proposed method can not achieve the best average, especially for some unsupervised methods. For the SOTA - Deep Shells, why do the proposed methods have the similar performance on Faust and Scape datasets instead of Shrec? Could there be some analysis and discussion of the results?

The description of the Prism decoder is not clear, and I cannot find the corresponding components in Figure 3. I suggest the author re-illustrate the figure and add more details for easier understanding.

Some mathematical notations are repeatable:
> F_i and F_j in eq3, F_1 and F_2 at line 1130


**Questions:**

1. For the Shape reconstruction module, why not directly use the vertex feature to predict the deformation on the vertex rather than using the features on the faces to predict the rotation of the faces? BTW, I think the optimization of eq. 5 is not trivial and needs more time to get optimal.

2. The proposed method is currently tailored towards complete shape correspondence. Have you thought of adapting it for partial shape-matching scenarios?

3. In the limitations section, you mentioned exploring ways to further expedite the training process. Could you provide more details on what you mean?

4. In the loss function, how do you determine the weight for each term and evaluate the importance of each term for the target?


**Limitations:**

The paper adequately addresses the limitations of the proposed method but could benefit from a detailed discussion of the potential negative societal impact. Besides, I think the running time should be evaluated. I think the optimization of Eq. 5 is too slow and takes other steps. The paper also mentions it in the last section.

---

> ### Author Rebuttal · Authors · 2023-08-06
>
> We thank the reviewer for their detailed and thoughtful comments and suggestions. We invite the reviewer to consider the general feedback that we provided to all reviewers in the "Author Rebuttal" answer. Below are our answers to the reviewer's questions.
>
> **Q1: Enhancing Comparisons with Related Work**
>
> We value your feedback. A visual comparison with three leading methods is in Figure 4 of the supplementary material, and quantitative comparisons across several well-known datasets [53, 25, 62, 35, 29, 77, 46] are provided in Tables 1 and 2 of our paper. We will be happy to relocate the visual comparison to the main content and add results from any other dataset you suggest, either during discussions or in the finalized manuscript.
>
>
>
> **Q2: Performance Comparison with Unsupervised Methods**
>
> We appreciate your observation. While our method may not top the average scores for Faust and Scape against certain unsupervised methods, these comparisons might not be apples-to-apples. The unsupervised methods train on extensive datasets for extended periods, sometimes for days [35, 36]. Our approach, in contrast, processes each shape pair without prior training, highlighting its zero-shot nature. Nevertheless, it still remains competitive with both supervised and unsupervised techniques and even achieves SOTA performance on the Shrec dataset.
>
>
> **Q3: Variability in DeepShells Performance Across Datasets**
>
> Thanks for noting this. DeepShells requires extensive training to exhibit proficient performance, as seen in its results on Faust and Scape datasets. However, Shrec is smaller, with 44 shapes just for evaluation. Usually, the best model from Faust or Scape is used for Shrec evaluation, as mentioned in several studies [26,70,50] and addressed in L263 of our paper. It's also crucial to recognize that DeepShells uses SHOT descriptors, which might be affected by remeshing changes [25, 70], a feature of the Shrec dataset. In contrast, our method evaluates each shape pair independently and is unaffected by prior data or remeshing.
>
>
> **Q4: Clarifying the Prism Decoder Illustration and Notation**
>
> Thanks for highlighting the ambiguities around the Prism decoder and repetitions in notation. In our revision, we will improve the Prism decoder illustration, capturing all details from Section 4.2, and include visuals for the latent code and the MLP's output. Additionally, we will address all repetitive notations for consistency across the paper.
>
>
> **Q5: Vertex vs. Face Features for Deformation Prediction**
>
> We appreciate your question about deformation prediction. We did a test using only DiffusionNet's output for direct vertex deformations, as detailed in the Supplementary Material's Table 4 (see the `DiffusionNet` row). Our results show that using the Prism decoder, which predicts each face's rotation and translation, offers about seven times better performance.
>
>
> **Q6: Addressing Concerns on the Optimization of Equation 5**
>
> We appreciate your feedback on Equation 5. To clarify, this equation is solved in closed form, as mentioned in L27 of the Supplementary Material. The matrix R is obtained from the SVD decomposition of a 3x3 matrix, which is very efficient.
>
>
> **Q7: Extending the Method to Handle Partial Shapes**
>
> We value your insight on partial shape correspondence. As mentioned, our method is geared towards complete shape correspondence. We believe that adapting to partial shape-matching is possible in principle, and for this, we would consider a strategy similar to DPFM [27]. This would require using the partial functional map estimation approach and potentially predicting a pointwise mask, indicating source shape regions aligning with the target, to adjust the MSE loss. This adaptation isn't straightforward, but a promising direction for future work.
>
>
> **Q8: Strategies to Accelerate the Optimization Process**
>
> We appreciate your query on optimization speed. As outlined in our limitations, we're actively seeking ways to expedite this:
>
>  - **Convergence Speed**: Adapting our learning rate or using a more advanced gradient descent method could hasten convergence. We used the Adam optimizer, but newer techniques like Adan optimizer [79] offer quicker convergence.
>
> - **Network Implementation**: With our networks (DiffusionNet and Prism-decoder's MLP) being MLP-based, a "fully fused" implementation [80] could significantly boost speed.
>
> We're optimistic that these changes can improve our optimization speed and plan to explore them in future work.
>
>
> **Q9: Determining and Evaluating the Weights of Loss Terms**
>
> Thanks for your question on loss term weights. As mentioned in L34 of the supplementary materials, we've uniformly weighted all loss components across datasets. Optimizing these weights could indeed enhance results.
>
>
> **Q10: Run Time Evaluation Concerns**
>
> We appreciate your input on run time evaluation. Please refer to Table 3 in the supplementary materials, where we've provided a detailed run-time assessment for our method.
>
>
> **Q11: Societal impact**
>
> Shape matching and analysis are crucial in fields like medical imaging, archaeology, and computational biology. The proposed methodology in this paper is highly applicable across these sectors, especially where data annotation is challenging or infeasible, and training datasets are scarce. Our approach reduces the need for extensive data labeling, aiding smaller research groups and leading to cost-efficient research. Moreover, it offers increased accuracy vital in areas like biology, where intricate shape analyses inform health conditions. While our method addresses a core issue in computer graphics and vision, and we anticipate mostly positive outcomes, it's essential to note the potential misuse in areas like surveillance. We strongly discourage such applications.

---

> > ### Comment · Reviewer_E6YV · 2023-08-16
> > **Reply**
> >
> > Thanks for the authors' detailed responses and clarifications. After reading the rebuttal and other reviews, most of the concerns have been fully addressed. I suggest all of the revisions should be presented in the revised paper. Besides that, I am happy to see some visual comparison with other baselines and the visual/numerical results on the diverse datasets.

---

> > > ### Comment · Reviewer_E6YV · 2023-08-20
> > > **Questions.**
> > >
> > > Could there be some updates for the new requests?

---

> > > > ### Author Response · Authors · 2023-08-20
> > > > **Addressing Feedback**
> > > >
> > > > We appreciate the reviewer's positive feedback. As highlighted in our previous response, we have provided numerical results from several datasets commonly referenced in the non-rigid shape-matching community [53, 25, 62, 35, 29, 77, 46]. We also plan to move the visual comparison, currently in Figure 4, into the main text of the revised paper. In response to the reviewer's final comment, we've updated our feedback for reviewer **nxSf**. It should be noted that all our new numerical and visual results for the rebuttal for other reviewers can be found in the PDF attached to the "Author Rebuttal" answer.

---

### Official Review · Reviewer_osbV · 2023-07-07

**Soundness:** 3 good
**Presentation:** 3 good
**Contribution:** 3 good
**Rating:** 7
**Confidence:** 3

**Summary:**

This paper works on learning a point-to-point mapping between two sets of deformable mesh vertices in a self-supervised manner. To this end, the authors extract features of two meshes from a DiffusionNet, solve for functional maps by an optimization problem, and finally convert the functional maps to a point-to-point map iteratively.

In the meantime, a per-face rigid transformation is generated from a shape decoder and is used to transform the source shape to the target shape. To demonstrate the effectiveness of the proposed method, the authors conduct experiments on both near-isometric and non-isometric datasets, and achieve reasonable performance as a self-supervised method.

**Strengths:**

- The paper is well organized and written. The problem is well defined in Sec.1, while the previous works and preliminary knowledge are also well introduced in Sec.2 and Sec.3.

- The two-stream (implicit shape transformation decoder and explicit functional map) pipeline is a reasonable design:
  - The low-rank functional map estimation is not only efficient but also facilitates the learning of feature extraction.
  - The MLP shape decoder and the PriMo energy regularize the predicted transformation by both network structure and training losses.

- As a self-supervised method, the performance is strong on both near-isometric and non-isometric data.

**Weaknesses:**

I do not see major weaknesses except for some details:
- The feature extractor should have more discussion, since it is key for the functional map prediction. Specifically, the author could either visualize or quantitatively measure the consistency of the extractor features. In addition, it would be better to explicitly indicate that the DiffusionNet is not pre-trained (in spite of lines 36~37) to avoid confusion with existing models.

- The pipeline figures could be better illustrated, such as the module blocks can be colored based on whether they are learned or have explicit formulation.

**Questions:**

The shape encoder might be redundant, and similar to DeepSDF the authors can directly learn the latent code jointly with other module parameters.

**Limitations:**

The limitations have been adequately addressed.

---

> ### Author Rebuttal · Authors · 2023-08-09
>
> We thank the reviewer for their detailed and thoughtful comments and suggestions. We invite the reviewer to consider the general feedback that we provided to all reviewers in the "Author Rebuttal" answer. Below are our answers to the reviewer's questions.
>
>
> **Q1: Further Insights on the Feature Extractor**
>
> Thank you for this question. Indeed, the features computed by our feature extractor are generally consistent with respect to the underlying maps. To illustrate this, we've included a visualization that showcases the consistency of the features generated by the feature extractor across a pair of shapes (See Figure 7 in the "Author Rebuttal" answer). In this figure, we reduced the dimension of the extracted features with PCA and then visualized them as RGB values. From this visualization, it's clear that the features from both shapes match well and emphasize the same areas. We will be happy to provide other similar illustrations in the final version.
>
>
> **Q2: Direct Latent Code Learning vs. Using an Encoder**
>
> We appreciate the reviewer's insightful observation. Indeed, we considered the choice of directly learning the latent code in a manner akin to DeepSDF. However, our empirical results indicated a superior performance when employing the encoder architecture. Specifically, for the Faust dataset, using a latent code gave a geodesic error of 4.0 versus our encoder's 1.9. For the Scape dataset, the errors were 9.9 and 4.7 respectively.
>
> A crucial distinction between our method and DeepSDF is the context of the optimization. Our SNK optimizes over a single shape pair in an **unsupervised** manner, whereas DeepSDF is trained on a substantial dataset using a supervised paradigm. This stronger training signal in DeepSDF can effectively assist in deriving robust latent codes. Moreover, our encoder offers the benefit of harnessing the input shape as an additional source of information, further enriched by the intrinsic regularization introduced by the network architecture  (as discussed in our response to reviewer **j61z**, Q3).
>
> We recognize the importance of this comparison and will ensure to integrate this ablation study in the revised version of the paper for clarity.
>
>
>
> **Q3: Exposition / Enhancements to the Pipeline Figure**
>
> We thank the reviewer for finding the paper "well organized and written". In our revised version, we will improve the visualization by color-coding the pipeline blocks to differentiate between optimized components, like the feature extractor, shape encoder, and parts of the decoder, compared to those with explicit formulation, such as the FM solver and the rotation and translation predictions in the decoder. Furthermore, to prevent any confusion, we will explicitly indicate that our approach *does not* rely on any pre-training, including the DiffusionNet feature extractor.

---

> > ### Comment · Reviewer_osbV · 2023-08-21
> > **Response to the authors**
> >
> > Dear authors:
> >   Thank you for the response.
> >   The rebuttal has addressed all my questions.

---

### Author Rebuttal · Authors · 2023-08-09

We thank the reviewers for their constructive comments. We find the suggestions to be beneficial for improving the quality of our work, making it clearer and more convincing.

Before responding to individual concerns, we stress the following contributions of our work:

 * We have devised a new decoder architecture rooted in the PriMo energy concept [**osbV**, **nxSf**]. This architecture facilitates the production of deformations that appear smooth and natural [**osbV**, **E6YV**, **nxSf**].

 * We have shown that a loss function, which imposes penalties on both spatial and spectral quantities [**osbV**, **j61z**, **nxSf**], is adequate for deriving matches on a single pair of shapes without any prerequisite training [**E6YV**].

 * We have developed a method for zero-shot shape matching that attains state-of-the-art results among methods that operate on individual pairs [**E6YV**, **nxSf**]. Furthermore, it competes with, and frequently outperforms, several supervised training-based approaches [**osbV**, **E6YV**].


We believe that all of the suggested changes can be easily done within a minor revision and we will make sure to address all of the comments and concerns in the final version. We will also release our code and data for full reproducibility of our results and to facilitate future work in this area.

---
In response to the reviewers' queries, we've added new references. For clarity, we'll enumerate them below.

[79] Adan: Adaptive Nesterov Momentum Algorithm for Faster Optimizing Deep Models. X. Xingyu et. al. 2022.

[80] Instant Neural Graphics Primitives with a Multiresolution Hash Encoding. T. Muller et. al. ToG 2022.

[81] Interactive curve-constrained functional maps. A. Gehre et. al. CGF 2018.

[82] Elastic Correspondence between Triangle Meshes. D. Ezuz et. al. CGF 2019.

[83] Deep Image Prior. D. Ulyanov et. al. CVPR 2018

[84] NCP: Neural Correspondence Prior. S. Attaiki et. al. Neurips 2022

[85] Dynamic Graph CNN for Learning on Point Clouds. Y. Wang et. al. TOG 2019

[86] Rethinking Network Design and Local Geometry in Point Cloud: A Simple Residual MLP Framework. X. Ma et. al. ICLR 2022

---

### Decision · Program_Chairs · 2023-09-21

**Decision:**

Accept (poster)

**Comment:**

This paper introduces a zero-shot approach for unsupervised learning of inter-shape correspondences. The key idea is to combine two state-of-the-art correspondence paradigms, i.e., the functional map framework which is intrinsic and the Primo deformation module which is extrinsic. Experimental results show that the approach combines the strength of both modules and achieves state-of-the-art results on common benchmark datasets. The submission received all positive reviews. The hybrid approach presented in this paper can stimulate future research. Please incorporate comments from the reviewers to prepare the final version.